# Application of Nano-Hydroxyapatite Derived from Oyster Shell in Fabricating Superhydrophobic Sponge for Efficient Oil/Water Separation

**DOI:** 10.3390/molecules26123703

**Published:** 2021-06-17

**Authors:** Chao Liu, Su-Hua Chen, Chi-Hao Yang-Zhou, Qiu-Gen Zhang, Ruby N. Michael

**Affiliations:** 1Key Laboratory of Jiangxi Province for Persistant Pollutants Control and Resources Recycle, Nanchang Hangkong University, Nanchang 330063, China; 1802077600006@stu.nchu.edu.cn (C.L.); 1902085229145@stu.nchu.edu.cn (C.-H.Y.-Z.); niatzqg@nchu.edu.cn (Q.-G.Z.); 2School of Engineering and Built Environment, Griffith University, Nathan, QLD 4111, Australia; ruby.michael@griffith.edu.au

**Keywords:** nano hydroxyapatite, superhydrophobic sponge, oyster shell, PDMS, oil/water separation

## Abstract

The exploration of nonhazardous nanoparticles to fabricate a template-driven superhydrophobic surface is of great ecological importance for oil/water separation in practice. In this work, nano-hydroxyapatite (nano-HAp) with good biocompatibility was easily developed from discarded oyster shells and well incorporated with polydimethylsiloxane (PDMS) to create a superhydrophobic surface on a polyurethane (PU) sponge using a facile solution–immersion method. The obtained nano-HAp coated PU (nano-HAp/PU) sponge exhibited both excellent oil/water selectivity with water contact angles of over 150° and higher absorption capacity for various organic solvents and oils than the original PU sponge, which can be assigned to the nano-HAp coating surface with rough microstructures. Moreover, the superhydrophobic nano-HAp/PU sponge was found to be mechanically stable with no obvious decrease of oil recovery capacity from water in 10 cycles. This work presented that the oyster shell could be a promising alternative to superhydrophobic coatings, which was not only beneficial to oil-containing wastewater treatment, but also favorable for sustainable aquaculture.

## 1. Introduction

The emergence and persistence of oil pollution in aquatic and marine ecosystems have raised universal health concerns on the environment and wildlife for decades. Many operational innovations introduced by the Maritime Agreement Regarding Oil Pollution (MARPOL) to regulate the discharge of oily water have contributed greatly to a noticeable decrease in marine pollution, whereas it is well recognized that a greater effort shall be carried out to treat the discharged oil considering the more stringent requirement on maximum permissible concentration (MPC) for oil worldwide [1,2,3]. For example, the MPC for oil in the bilge water discharges through the oily water separator is the well-known 15 ppm standard enforced by MARPOL since 1983 [4,5]. Therefore, oil/water separation methods, including absorption [6], flocculation [7], and electroflocculation [8], have recently been applied to purify oily water and adopted as an alternative to biological treatment for its time-saving property [9]. Of these, absorption appears to be more attractive due to its universal function in removing all forms of oil from the oily water, high oil recovery efficiency, and convenient post-treatment.

Usually, the absorbents used for the treatment of oily water are either natural products or close derivatives, including plant fibers [10], wool [11], organophilic clay [12], exfoliated graphite [13], starch [14], and cellulose-based materials [15], which possess eco-friendliness, low cost, and easy accessibility. However, these materials exhibited low selectivity for oil and water, inadequate buoyancy, and poor mechanical properties, resulting in a low recovery rate of oil and inconvenient recycling of absorbent [16]. Hence, the exploration of new oil absorbents exhibiting high oil/water selectivity, low density, and excellent recyclability is of great economic and ecological significance for oil pollution prevention.

Superhydrophobic materials with different wetting properties between water and oil are promising to effectively separate oil from water. Many superhydrophobic materials have been fabricated for oil/water separation by using metallic mesh [17], glass [18], textile [19], and sponge [20] as substrates. Compared with other substrates, sponges are more portable and convenient to recycle owing to their low density, high flexibility, and three-dimensional architectures. In addition, high porosity and large surface area endow sponge with high absorption capacity. Additionally, sponge generally contains hydrophilic groups, such as -OH and -COOH, which offer potential for further modification [21]. Therefore, many works have been devoted to preparing superhydrophobic sponge for efficient oil/water, such as melamine sponge [22], polyurethane sponge (PU sponge) [21], and cellulose sponge [23], of which PU sponge has been considered as an ideal substrate due to its low cost, reproducibility, and compatibility [20].

Generally, the superhydrophobic surface can be acquired by introducing rough structures and polymer chemicals of low surface energy [24]. For example, polydimethylsioxane (PDMS) and poly(vinylidene fluoride) (PVDF) were introduced to reduce the surface energy and thus increase the hydrophobicity of nanoparticle surfaces [18,22]. Considering the toxicity of fluorinated polymers, the employment of PDMS to design and construct superhydrophobic surfaces is a better option. In addition, metals and their oxides have been widely employed to develop micro/nanostructures to increase surface roughness [25]. To some extent, the metal-derived superhydrophobic surfaces exhibited satisfactory oil/water separation. However, there are still some problems existing that seriously restrict their practical applications. First, most of the prepared micro/nanostructures of superhydrophobic surfaces are too fragile to endure mechanical forces or chemical erosion [24]. Second, the release of metal ions and nanoparticles from the metal-containing superhydrophobic material may induce toxicity to the aquatic ecosystem and human health. Therefore, it is of great significance to develop biocompatible nanomaterials with fine mechanical properties. Interestingly, hydroxyapatite (HAp) nanoparticle has been widely studied in the medical area because of its superior biocompatibility and specific bioactivity for drug, bone mineral, and gene delivery [26]. For instance, hydroxyapatite has been exploited as an orthopedic and dental implant due to its mechanical, thermal, and chemical stability [27]. HAp can be easily prepared from various natural materials, including biogenetic calcium carbonate (CaCO_3_) and mineral-derived calcium compounds [28].

In recent years, shellfish aquaculture has achieved large development to cover the large consumption of seafood products. A large number of oyster shells that have been seriously undervalued have become waste or low-value resources, occupying some tidal flats and land, corrupting and smelling, and bringing many adverse effects to the environment [29]. Therefore, the comprehensive development and utilization of oyster shells are of great significance. How to effectively develop and utilize oyster shell resources and turn them into treasures is the research purpose of this subject.

Herein, for the first time, we prepared green-based PU sponge (EP/SPU) by using hydroxyapatite prepared from oyster shells and polydimethylsiloxane (PDMS). There are few reports in the literature about the use of shell/PU sponge for oil/water separation. Of course, some researchers used oyster shells to prepare a superhydrophobic foam; however, because the main component of oyster shells is calcium carbonate, the prepared materials are easily corroded by strong acids and lose their function. It is particularly important to explore a material that can use oyster shells to prepare acid and alkali resistance. In this study, we used oyster shells to prepare a nanomaterial that can better resist acid and alkali corrosion so that the prepared superhydrophobic sponge has stronger acid and alkali resistance and can operate stably in the pH range of 4–12. This not only realizes the resource utilization of oyster shells, but also increases the possibility of superhydrophobic sponges in oil–water separation applications.

Recycling and reuse of discarded oyster shell can considerably increase the profit of oyster farming and achieve sustainable development. Herein, this study aims to explore the possibility of oyster shell in preparing Hap-coated superhydrophobic material.

## 2. Materials and Methods

### 2.1. Materials

The discarded shells of the species *Crassostrea angulata* were collected from a local seafood market in Nanchang, China. The shells were treated with ultrasonic cleaning, air-drying at room temperature, mechanically crushing and milling to fine powder, and sieving with a 200-mesh sieve to obtain the oyster shell powders in sequence.

The polyurethane (PU) sponge with an average pore size of 200 μm was supplied by SuQiang Aitao Trading Co., Ltd. (Jiangsu, China). Hydroxyl-terminated polydimethylsiloxane (PDMS, silicone oil) used as the coating material was acquired from Aladdin Reagent Co., Ltd. (Shanghai, China). Hexane, toluene, and (NH_4_)_2_HPO_4_ were purchased from Shantou Xilong Chemical Co., Ltd. (Guangdong, China). Oils including soybean oil and lube oil were commercial products in a local supermarket. Gasoline, as well as diesel, was kindly provided by a local gas station. Hydrochloric acid and sodium hydroxide were acquired from Aladdin Reagent Co., Ltd. (Shanghai, China). All materials were used as received if not stated otherwise.

### 2.2. Preparation of HAp Coated Polyurethane Sponge (HAp/PU Sponge)

HAp was synthesized in a way similar to that originally reported by Yang et al. [25]. Briefly, 2.00 g of the oyster shell powders was added to a solution obtained by dissolving 3.17 g of (NH_4_)_2_HPO_4_ with 50 mL deionized water in a Teflon bottle, and the mixture was vigorously stirred to achieve homogeneous suspension. The bottle was then sealed in a stainless steel autoclave, which was kept in a drying oven at 220 °C for 6 h. After the autoclave cooled to room temperature, the precipitates were collected and milled in a mortar to obtain a fine HAp powder.

Subsequently, HAp/PU sponge was developed according to the procedure previously described [30] with some modification. Typically, a dilute solution of hydroxyl-terminated polydimethylsiloxane (PDMS) in hexane (1:20, *v*/*v*) was prepared by magnetic stirring for 30 min. A ceramic slurry was prepared by dispersing 0.5 g of the as-prepared hydroxyapatite powder in 40 mL of the dilute solution. PU sponges were cut to an appropriate size (20 × 20 × 15 mm) and then completely immersed in the HAp slurry for 15 min. The PU sponge was squeezed to remove excess slurry and dried at 60 °C for 12 h. For comparison, the PU sponge was directly immersed into the dilute solution without HAp powder addition to synthesize PDMS coated PU (PDMS/PU) sponge under the same experimental conditions as above.

### 2.3. Physiochemical Characterization

Evaluation of structural and chemical properties of the samples was carried out by various characterization techniques. We cut the sponge into 2 mm thick slices and sprayed gold on the surface, Morphologies of the obtained HAp, PU sponge, PDMS/PU sponge, and HAp/PU sponge were observed using KYKY-EM3900M scanning electron microscopy (SEM) with an accelerating voltage of 20 kV. The crystal phase of HAp power and HAp coated on the PU sponge was analyzed using a Bruker D8 Advance X-ray diffractometer with Cu Kα radiation (λ = 0.15406) at a scanning rate of 8 °/min in the 2θ ranging from 10° to 70°. Fourier transfrom infrared spectra (FT-IR) were measured on a Nexus 670 FT-IR spectrometer (VERTEX 70, Bruker, Ettlingen, Germany). The surface element composition of the HAp/PU sponge was analyzed by an Axis Ultra DLD system from Kratos with a resolution of less than 0.2 eV. The water contact angle (WCA) was measured on an optical contact angle meter system (SDC-100, Sindin, Guangdong, China).

### 2.4. Adsorption Performance Test and Stability Investigation

The oils or organic solvents, including soybean oil, diesel, lube, toluene, and chloroform, were used in the absorption test. Firstly, a piece of as-prepared nano-HAp/PU sponge was immersed in 10 mL of the above-mentioned solvent until it was fully saturated. The saturated sponge was then picked up and quickly weighed to avoid evaporation of the absorbed solvent. The mass-based absorption capacity (*Q_m_*) was calculated as follows:(1)$$Q_{m}=\frac{M_{t}-M_{i}}{M_{i}}$$
where the *M_t_* and *M_i_* were the weight of the nano-HAp/PU sponge before and after the absorption experiment, respectively. 

The cyclic absorption performance of the nano-HAp/PU sponge was evaluated by the absorbed capacity of the above-mentioned solvent using the absorption–extrusion method. The absorbed capacity in cycle n (*Q_n_*) was calculated by the following equation.
(2)$$Q_{n}=\frac{M_{n,t}-M_{i}}{M_{i}}$$
where *M_i_* and *M_n_*_,t_ are the is the initial mass of the dry sponge and total mass of the nano-HAp/PU sponge in cycle n absorption.

In order to evaluate the chemical stability of fabricated superhydrophobic surfaces on the sponge, WCA measurement was carried out after nano-HAp/PU sponge samples were immersed in the hydrochloric acid aqueous solution (pH = 3), neutral solution (pH = 7), and sodium hydroxide aqueous solution (pH = 11) for 1 h to test their corrosion resistance. In order to evaluate the impact of HAP loading on material properties, we prepared ceramic slurry by dispersing 0, 0.2, 0.5, 0.8, 1, and 1.2 g HAP powder in 40 mL dilute solution during the preparation of superhydrophobic sponge and perform WCA measurement and observation on the above sponge.

### 2.5. Oil−Water Separation

The performance of nano-HAp/PU sponge in continuous oil/water separations was tested with a special separation system consisting of a rubber hose, two conical flasks, and a peristaltic pump. As shown in Figure 1, one end of a rubber hose was firmly attached to the functionalized nano-HAp/PU sponge, which was dipped into a conical flask containing 250 mL of diesel–water mixture (1:2, *v*/*v*); the other end was dropped into the other conical flask to collect oil. The oil started to outflow through the functional sponge, driven by the pressure of the pump. The whole process was recorded with a digital camera.

## 3. Results and Discussion

### 3.1. Chemical Composition Analysis

Hydroxyapatite obtained from oyster shells and HAp-coated PU sponge were first characterized by X-ray diffraction (XRD) analysis. As shown in Figure 2, two peaks centered at about 31° and 27° can be corresponded to the XRD patterns of pristine sponge. Additionally, several sharp peaks at 25.8°, 28.1°, 28.9°, 32.2°, 32.9°, 34.1°, and 39.8° in Figure 2a can be attributed to the (002), (102), (210) (112), (300), (212), and (310) planes of pure hexagonal HAp (JCPDS card No. 09-0432) [25]. These results suggest that the well-prepared HAp particles were successfully loaded on the PU sponge.

The surfaces of the PU sponge before and after HAp loading were characterized by attenuated total reflectance (ATR)-FTIR to further confirm the structural and chemical changes of the samples. The pristine PU sponge exhibited typical characteristic peaks of 3726, 2879, 1743, 1616, 1103, and 632 cm^−1^, which are related to the stretching vibrations of O-H, C-H, C=O, N-H, epoxy C-O, and =C-H groups, respectively (Figure 3b). After modifying the PU sponge with the HAp particles, several characterized peaks at 1037, 632, and 613 cm^−1^ can be clearly observed for the HAp/PU sponge (Figure 3c). These peaks are associated with the bending vibration of P-O and the stretching vibrations of P-O and P=O groups present in HAp, respectively (Figure 3a), which are close to the vibrations observed by Shaly et al. [31]. Moreover, two new distinct peaks appeared at 806 and 1265 cm^−1^, possibly ascribing to the symmetrical stretching vibration of Si-O-Si groups and the symmetrical bending vibration of Si-CH_3_, respectively, which revealed the effect of cross-linked PDMS on surface modification. Under the alkali condition, the alkoxy groups of siloxanes molecules can be hydrolytically accelerated to form silanol groups, which can covalently react with the hydroxyl groups of HAp at a low temperature of 80 °C [32]. Thus, these FTIR results clearly demonstrate the successful covalent coating of silane molecules with HAp molecules to the PU sponge surface.

### 3.2. Surface Morphology Analysis

The sharp three-dimensional images of HAps, original sponge, and HAp-coated PU sponge, provided by FE-SEM at different magnifications, shed light on the topography and morphology of the samples under study. It can be seen from Figure 4c that the un-processed PU sponge has a 3D porous network structure with a pore size ranging from 200 to 600 μm, which is very helpful for the loading of nanoparticles and is essential for maintaining high adsorption capacity [33]. Magnified images of the unprocessed PU sponge (Figure 4c,d) present the smooth skeleton surface, which has no micro-nano structure on its surface and cannot achieve superhydrophobicity. After PDMS modification and HAp loading, the 3D porous framework of the PU sponge was still maintained (Figure 4e), indicating that the porous skeleton structure was not destroyed during the solution–immersion. In addition, the dense hydroxyapatite particle was randomly distributed on the PU sponge surface (Figure 4f), rendering the micron-scale roughness of the coated surface (Figure 4g). It has been shown that the special wetting behavior of the surface is closely related to its morphology and chemical composition so that the combination of high surface roughness and lower surface energy is essential for obtaining a superhydrophobic surface [34]. Both the enhanced surface roughness of PU by micro-sized and nano-sized hydroxyapatite (Figure 4a,b) and the lower surface energy led by PDMS give the potential to achieve improved superhydrophobicity of PU.

The spectra obtained from the energy dispersive spectrometer (EDS) analysis test for hydroxyapatite, PU sponge, and nano-HAp/PU sponge are exhibited in Figure 5. The EDS spectrum shows the presence of calcium, phosphorous, oxygen, and silicon elements of the nano-Hap/PU sponge, indicating the successful loading of HAp on the sponge and the effectiveness of the preparation method. Moreover, the emerging Si peak indicates the successful modification of PU sponge by PDMS, which is beneficial to lowering the surface energy.

### 3.3. Wetting Performance of the Superhydrophobic Sponge

The water contact angle (WCA) is often used to measure the wettability of the modified surface, which can reflect the fluid transport selectivity during oil/water separation. It has been widely accepted that superhydrophobicity (WCA > 150°) is normally needed to realize high selectivity [30]. The water contact angle values of different samples were shown in Figure 6. As shown in Figure 6, after the modification of PDMS, the water contact angle of the PU sponge slightly increased from 117° to 133°, indicating that the enrichment of PDMS on the surface could improve hydrophobicity. PDMS is supposed to decrease the surface energy and increase the surface roughness [22]. Moreover, due to the low cost, easy fabrication, high durability and flexibility, and self-healing properties, PDMS was extensively used for the preparation of new superhydrophobic materials with potential applications in phase separation [18]. However, PDMS modification alone presented a negligible effect on surface roughness, which is sufficient to obtain high superhydrophobicity. Fortunately, further introduction of HAp particles largely enhanced the surface roughness, and thus the water contact angle of the nano-HAp/PU sponge increased to above 150°. In this case, PDMS acted as both a low surface energy modifier and an adhesive. As a result, the introduction of PDMS and HAp obviously changed the wettability of the PU sponge. A single water droplet depositing on the HAp/PU sponge surface presented almost a perfect sphere (Figure 6c), while it exhibited a hemisphere on the surface of the pristine sponge (Figure 6a). Due to the as-formed micro/nanostructures on the HAp/PU sponge surface, air was trapped among these hierarchical rough structures so that it possessed high superhydrophobicity of water rather than oil (Figure 6d) [33]. Thus, the nano-HAp/PU sponge showed a high static water contact angle and a low oil contact angle.

Corrosion resistance of superhydrophobic materials is a crucial factor for their practical applications. Previous studies showed that the inert properties of microstructural constituents greatly contributed to the chemical stability of their bulk materials [24]. Indeed, the change of the microstructure on the HAp/PU surface under acid stress could influence the stabilization and wetting behavior of the PU sponge, which strongly correlated with its efficiency in practical oil–water mixture separation [26]. Thus, acid-induced behavior of water contact angles was investigated by immersing the as-prepared nano-HAp/PU sponges in simulated water at various pH for 1 h.

It is encouraging that the HAp/PU sponge could remain superhydrophobic within the pH range of 2–11 (Figure 7). In a strong alkaline solution, the WCA of the nano-HAp/PU sponges appeared similar as under neutral conditions, which is ascribed to the inherent stability of HAp. After being immersed in strong acidic solution, its WCA was observed to undergo a gradual change by 5°, almost retaining superhydrophobicity. The slight decrease may be due to the tiny H+ absorbed on the surface of HAp/PU sponge by electrostatic interaction.

Therefore, the good chemical stability of the HAp/PU sponge can be attributed to the chemical inertness of HAp and the strong covalent bonding among HAp, PMDS, and PU sponge. Thus, the HAp/PU sponge could not be damaged, even under strong acidic and alkaline solution conditions.

In order to study the effect of HAp loading on the performance of superhydrophobic sponges, we added HAp of different qualities during the sponge preparation process and measured their water contact angles. As shown in Figure 7, the WCA of the sponge modified by PDMS alone is 130.0°, and the water contact angles of the sponges with 0.2, 0.5, 0.8, 1.0, and 1.2 g HAp added are 144.1, 156.0, 146.8, 143.0, and 141.2, respectively, indicating that the loading of HAp increases the sponge’s wetting properties, playing a key role in the preparation of superhydrophobic sponges. With the increase of HAp dosage, the water contact angle of the modified sponge is gradually increasing. This is because the increase of HAp increases the rough structure of the sponge surface and makes the sponge more hydrophobic. When the dosage of HAp is 0.5 g, the hydrophobic performance of the sponge is the best, and its water contact angle is greater than 150°. When the amount of HAp added is greater than 0.5 and gradually increases, the water contact angle of the modified sponge gradually decreases, indicating that the load of excessive HAp particles may cause the surface of the sponge to be covered by the particles in a large area, which is not conducive to the construction of rough surfaces, which reduces the water contact angle of the sponge.

### 3.4. Performance of Evaluation of Nano-HAp/PU Sponge as Oil Absorbent

Herein, diesel and chloroform were used to study the selective absorption of oil above and below the water, respectively. As presented in Figure 8a, once the dyed diesel floating on the surface of the water contacted the modified sponge, it was rapidly absorbed into the skeleton of the sponge. Interestingly, when the nano-HAp/PU sponge was transferred to the water/chloroform mixture, it removed chloroform droplets within 5 s without any obvious residue (Figure 8b and Appendix A).

The nano-HAp coated on the skeleton can be fully wetted by contacted oil, and these oils quickly diffused into the sponge with the aid of capillary force, which rendered the modified sponge both of superhydrophobicity and superoleophilicity properties in water. Subsequently, the absorbed oil without apparent water was collected through simple squeezing, which means a fast, economic, and effective separation process. Therefore, the maximum adsorption capacity of oil by nano-HAp/PU sponge was further evaluated to test its potential in practical cleanup of organic pollutants from water. In a typical adsorption measurement, soybean oil, diesel, lube, toluene, and chloroform were chosen as adsorbates. The nano-HAp/PU sponge had a wide range of absorption capacities for different adsorbates depending on the viscosity and density of the oil or organic solvents, of which the adsorption capacity for soybean oil, diesel, lube, toluene, and chloroform can reach 11.3, 9.8, 11.4, 17.1, and 22.7 g.g^−1^, respectively (Figure 9a). In particular, the absorption capacity for chloroform is almost 23 times that of the nano-HAp/PU mass itself, which demonstrates the excellent absorption capacity of nano-HAp/PU sponge for oil/organic solvents as well as other reported oil absorbents [18,22].

In addition, nano-HAp/PU could maintain superhydrophobic and stable absorption capability for organics of either high or low viscosity after cycling for even 10 cycles (see in Figure 8b). A slight fluctuation of oil-absorption capacity along the cycling test could be observed, which may be due to the residual oil remaining in the pores of the sponge. After oil extrusion, the HAp/PU sponge could quickly recover its original shape without any deformation due to the excellent compressibility and elasticity of the PU sponge. Additionally, the microstructure of HAp composition was not damaged heavily even undergoing cyclic extrusion, indicating the high stability under mechanical stress. The good mechanical stability of HAp/PU sponge is beneficial to ultrawetting materials for practical oil/water separation.

### 3.5. In Situ Separation of Oil/Water Mixtures

Furthermore, the continuous separation performance of oil/water mixture with superhydrophobic nano-HAp/PU sponge was investigated by running a pump-driven system (Figure 10). The separation performance of oil from water with pristine PU sponge was also evaluated by the same method for comparison. Once the pump works, the diesel dyed with oil red was quickly and continuously passed through the fixed nano-HAp/PU sponge and was then collected into the right glass conical flask. Appendix A showed that the superhydrophobic sponge could recover 83 mL of dyed diesel from 250 mL of the static and immiscible oil/water mixture within 50 s, while the water level in the left glass conical flask remained unchanged. The separation efficiency was calculated as 99.6% by using the volume ratio between the collected diesel oil and that initially added in the mixture. In contrast, the unmodified sponge could not effectively collect diesel from the oil/water mixture, even connected to pump-assisted separation device (see in Appendix A). These results indicated that the presence of nano-HAp endowed the HAp/PU sponge with the excellent oil–water separation ability, which is consistent with the results of the wetting performance discussed in Section 3.3.

## 4. Conclusions

Apparently, abundant biogenetic CaCO_3_ with low cost is an attractive raw material for HAp preparation. Specifically, oyster shell was extensively used to synthesize HAp nanoparticle via hydrothermal treatment [35]. Therefore, oyster shell could be a promising raw material for fabricating a HAp-nanoparticle-coated superhydrophobic surface. A kind of nano-hydroxyapatite material was developed from discarded oyster shell via a simple and low-cost procedure, which was then coated on the surface of pristine sponge. The uniformly distributed nano-HAp along the PU skeleton formed a microstructure that changed the wetting properties of PU sponge from hydrophobicity to superhydrophobicity. The resultant nano-HAp/PU sponge is not only an adsorbent that can adsorb oil or organics, but also an excellent separation material that can continuously separate oil from water. It is a promising candidate for oil/water separation method for its high absorption capacity, low cost, and eco-friendliness. Additionally, some functional groups such as -NH_2_ and -OH on sponge provided a chemical connection between the HAp coating and PU substrate, which improved the durability of the surface and reusability of the HAp/PU sponge. The nano-HAp/PU sponge can be used in combination with the pump to deal with an oil spill emergency by recovering oil from water. In addition, the superhydrophobic coating on PU sponge is derived from discarded oyster shell, which provides a novel strategy to solve the serious pollution problem caused by accumulation of oyster shell. These advantages make nano-HAp/PU sponge a potential separator candidate for the recovery of oil from water for environmental protection application.

## Figures and Tables

**Figure 1 molecules-26-03703-f001:**
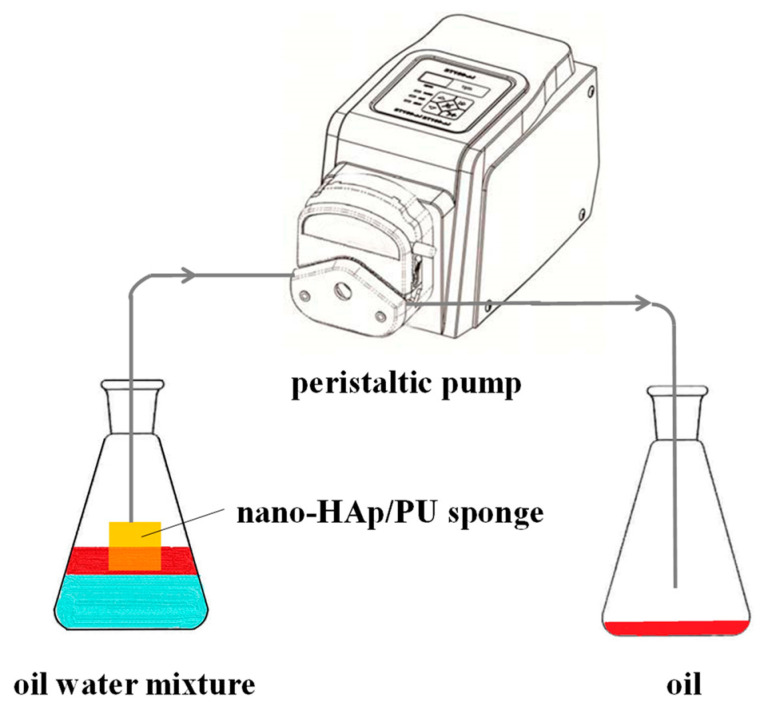
Scheme of the continuous in-situ oil/water separation.

**Figure 2 molecules-26-03703-f002:**
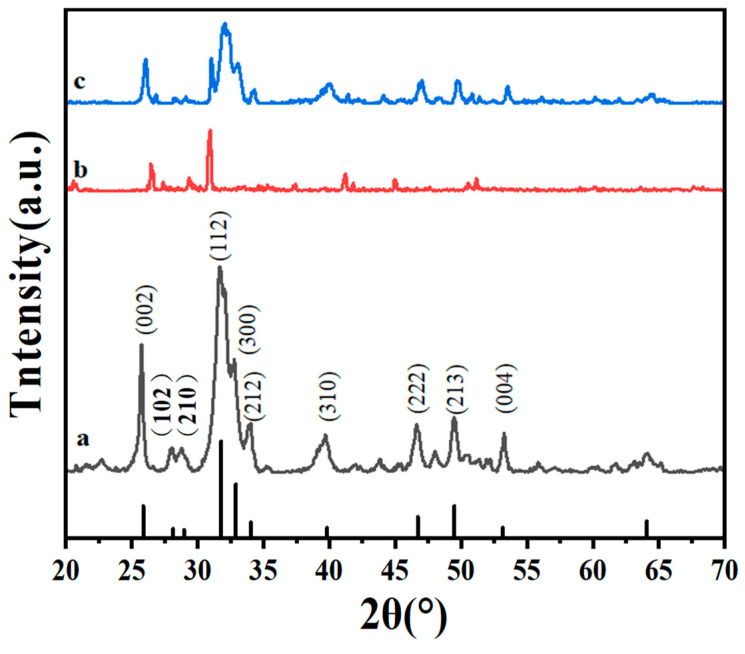
XRD patterns of (**a**) nano-HAp, (**b**) PU sponge, and (**c**) nano-HAp/PU sponge.

**Figure 3 molecules-26-03703-f003:**
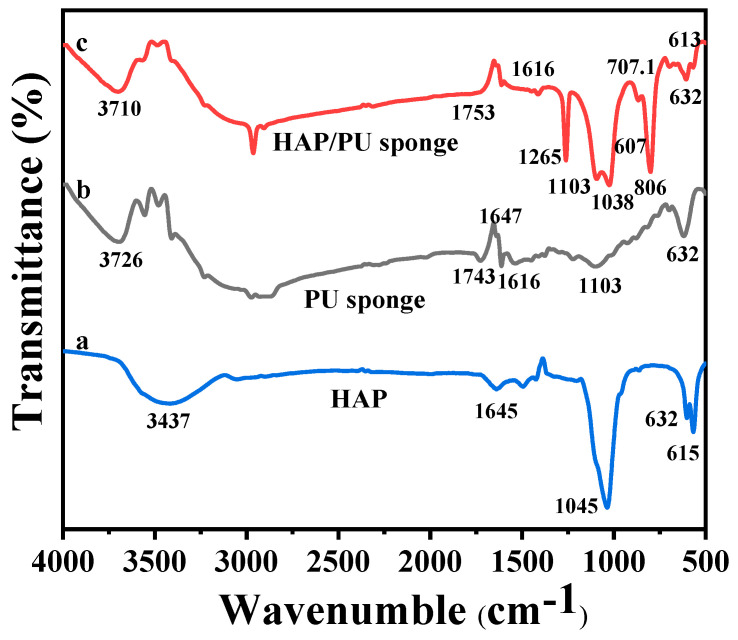
FTIR spectra of (**a**) nano-HAp, (**b**) PU sponge, and (**c**) nano-HAp/PU sponge.

**Figure 4 molecules-26-03703-f004:**
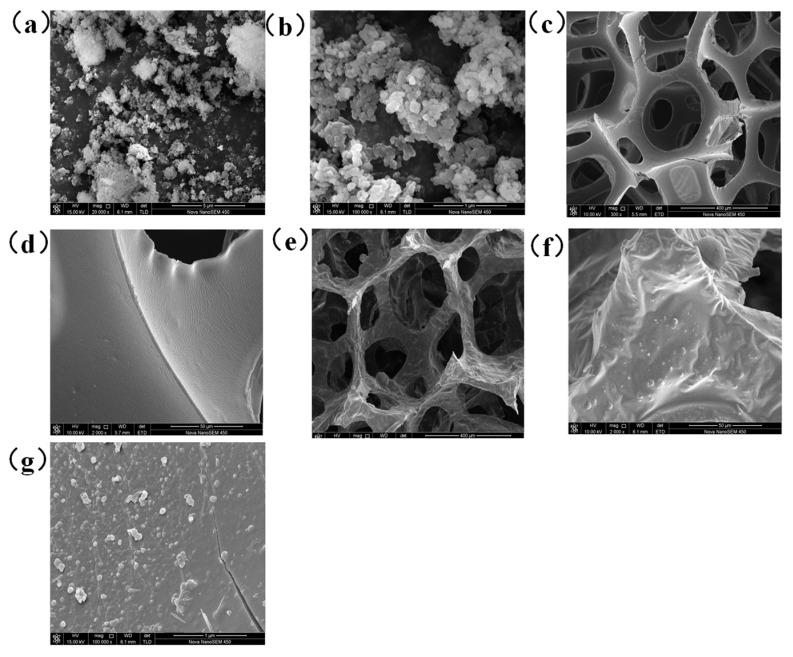
SEM images of nano-HAp (**a**,**b**), pristine PU sponge (**c**,**d**) and nano-HAp/PU sponge (**e**–**g**).

**Figure 5 molecules-26-03703-f005:**
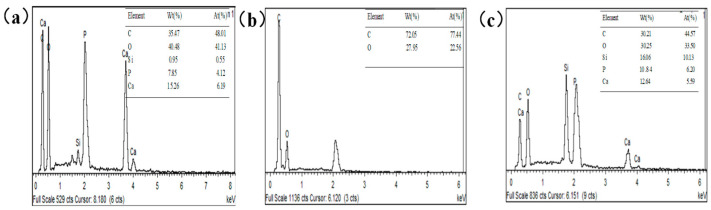
EDS images of (**a**) nano-HAp, (**b**) pristine PU sponge, and (**c**) nano-HAp/PU sponge.

**Figure 6 molecules-26-03703-f006:**
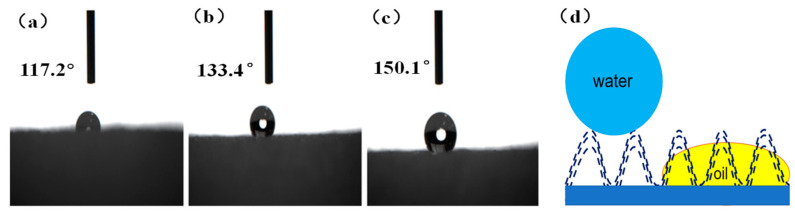
The water contact angle of (**a**) pristine PU sponge, (**b**) PU sponge with PDMS and without nano-Hap, (**c**) nano-HAp/PU sponge, and (**d**) superhydrophobic and superhydrophilic model.

**Figure 7 molecules-26-03703-f007:**
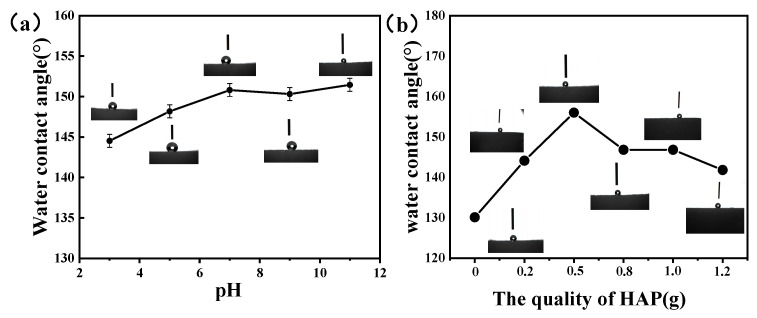
Change in the water contact angle of the nano-HAp/PU sponge after treatment at different pH values (**a**) and quality of HAP (**b**) added. Schematic illustration of the superhydrophobicity and superoleophilicity of the nano-HAp/PU sponge.

**Figure 8 molecules-26-03703-f008:**
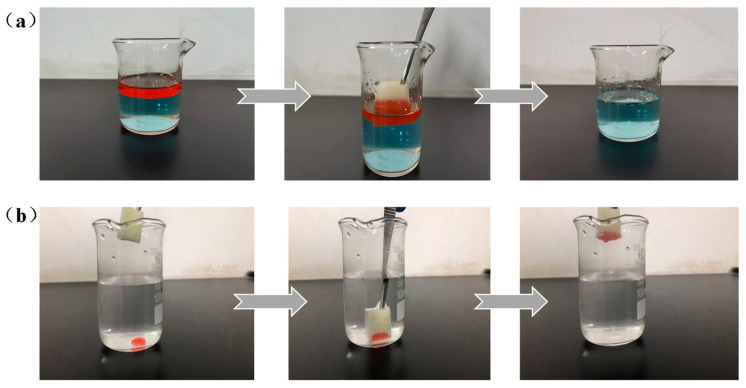
Selective oil absorption performance of the nano-HAp/PU sponge: (**a**) removal of diesel (dyed with oil red) below the water. (**b**) Removal of chloroform (dyed with oil red) below the water with nano-HAp/PU sponge.

**Figure 9 molecules-26-03703-f009:**
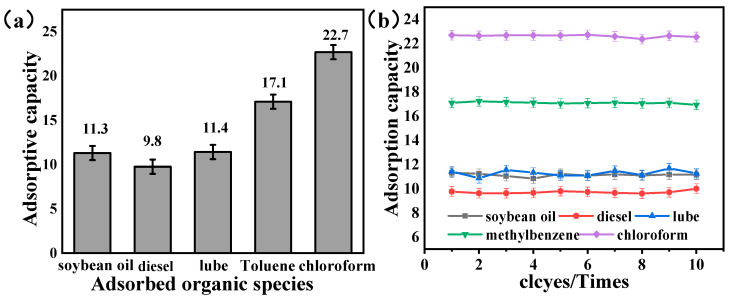
The nano-HAp/PU sponge maximum absorption for variety of organics (**a**) initially; (**b**) in absorption cycle effect.

**Figure 10 molecules-26-03703-f010:**
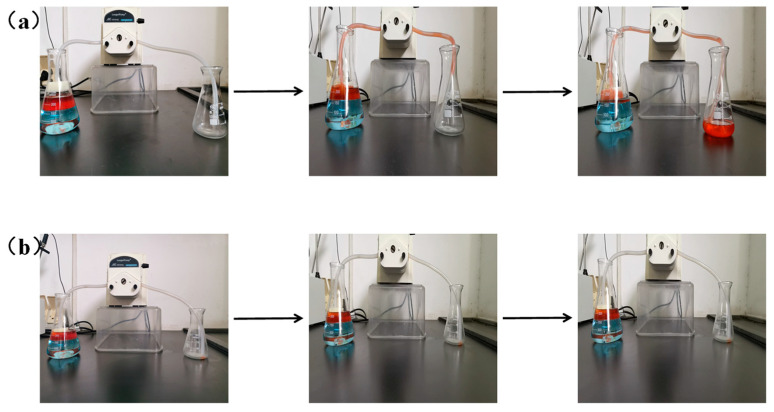
Separation of a mixture of diesel and water by the nano-HAp/PU sponge (**a**) and PU sponge (**b**).

## Data Availability

The data used to support the findings of this study are available from the corresponding author upon request.

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
