# Peer review of "Application of Nano-Hydroxyapatite Derived from Oyster Shell in Fabricating Superhydrophobic Sponge for Efficient Oil/Water Separation"

_molecules, 2021, doi:10.3390/molecules26123703_

Round 1
Reviewer 1 Report
In this paper, the application of nano-hydroxyapatite derived from oyster shell in fabricating superhydrophobic sponge for efficient oil/water separation is demonstrated. The following concerns or comments should be addressed properly.
- The introduction part is somewhat redundant. Regarding citations, the authors can assign them better to the discussion part, the expression will be more compact.
- In the Materials and Methods part, acid and base being used must be described.
- In Fig. 2, two peaks in the 2θ range from 25 to 30 degree must be assigned. Proper scales on the horizontal axis must be arranged to indicate the 2θ angles. From these patterns, I don’t agree with the good crystallinity of HAp. The crystallite size must be quantitatively discussed if the authors propose good crystallinity as has been stated.
- In Fig. 3, the wavenumber 3642 cm^-1 is indicated inside. However, the position of the band center is inconsistent to the wavenumber. The wavenumber in the figure must be corrected. Additionally, the assignment to correct vibration mode must be described also in the result.
- The loaded amount of HAp in the coating is important. However, the amount is not indicated anywhere. Therefore, at this present stage, it is impossible to evaluate the contribution of HAp in various properties of this coating, as compared to the PDMS one as a reference.
- The discussions are rather qualitative, for example, “high surface roughness” with no evidence, including SEM images.
- The magnification indicators in Fig. 4 are unreadable. Sample pre-treatment, for example Au coating, did not be needed to obtain the images?
- The vertical-to-horizontal ratio of figure is incorrect all in Figs. 5-10. It is hard to understand the phenomena exactly.
- In conclusion, the authors claim “uniformly” distributed HAp along the PU skeleton. What is the evidence of uniformity?
Reviewer 2 Report
It is a good article with clear objectives, proper range of investigations and conclusions supported by the results. I recommend only minor revision:
line 41-44: Among natural sorbents starch should be mentioned here:
“Usually, the absorbents used for the treatment of oily water are either natural products or close derivatives, including plant fibers [10], wool [11], organophilic clay [12], exfoliated graphite [13], starch [13a] and cellulose-based materials [14], which possess eco-friendliness, low cost, and easy accessibility.”
13a. Kugler, S.; Spychaj, T.; Wilpiszewska, K.; GorÄ…cy, K.; Lendzion-BieluÅ„, Z. Starch-Graft Copolymers of N-Vinylformamide and Acrylamide Modified with Montmorillonite Manufactured by Reactive Extrusion. J. Appl. Polym. Sci. 2013, 127, 2847–2854, doi:10.1002/app.37630.
line 97: unnecessary space
Figures: all figures have too low resolution, they are blurry as "oil in the water". Please, improve this; maybe the use of .EMF format for the pictures can solve this issue. Furthermore, ensure, that also the pictures are not forced to be compressed in the options of your Word application.
All charts should be formatted similarilly, major ticks, minor ticks, fonts (normal/bold) should be uniformed (eg. Figs 2, 3, 5, 7, 9).
Figure 4: the scale of all sub-figures must be identical and, at least, clearly defined in all 6 pictures.
line 234: unnecessary space (a, d)
Fig 5. is not only blurry, but also misses its proportions, as well as Fig. 7.
Round 2
Reviewer 1 Report
I understand that the author made an effort to revise the manuscript. However, (1) the source of the acid and base is not still described, (2) the scale indicators needed to understand the SEM images are still not provided, and (3) all the figure proportions after Fig. 5 have not been corrected. The amounts of used HAp to fabricate the composites were fortunately presented, in order to discuss their properties quantitatively. But the non-linearity for the hydrophobicity of the coating to the amounts of used HAp has not been discussed at all.
